# Use of Probiotics During Antibiotic Therapy in Pediatrics: A Cross-Sectional Survey of Italian Primary Care Pediatricians

**DOI:** 10.3390/antibiotics14060577

**Published:** 2025-06-04

**Authors:** Giacomo Biasucci, Maria Elena Capra, Antonella Giudice, Delia Monopoli, Brigida Stanyevic, Roberta Rotondo, Alessandro Mucci, Cosimo Neglia, Beatrice Campana, Susanna Esposito

**Affiliations:** 1Pediatrics and Neonatology Unit, Guglielmo da Saliceto Hospital, 29121 Piacenza, Italy; giacomo.biasucci@unipr.it (G.B.); m.capra@ausl.pc.it (M.E.C.); 2Department of Medicine and Surgery, University of Parma, 43126 Parma, Italy; antonella.giudice@unipr.it (A.G.); delia.monopoli@unipr.it (D.M.); brigida.stanyevic@unipr.it (B.S.); roberta.rotondo@unipr.it (R.R.); alessandro.mucci@studenti.unipr.it (A.M.); negliamino@gmail.com (C.N.); beatricerita.campana@unipr.it (B.C.); 3Pediatric Clinic, University Hospital of Parma, 43126 Parma, Italy

**Keywords:** antibiotics, antibiotic-associated diarrhea, antimicrobial stewardship, microbiota, probiotics

## Abstract

**Background:** Antibiotic-associated diarrhea (AAD) is a common adverse effect of pediatric antibiotic therapy, often linked to gut microbiota disruption. Probiotics may help prevent AAD when appropriately selected and dosed. **Methods**: We conducted a cross-sectional survey to assess the attitudes and prescribing habits of Italian Primary Care Pediatricians (PCPs) regarding the use of probiotics during antibiotic treatment. A digital questionnaire comprising 23 mandatory multiple-choice items was distributed to 980 PCPs across Italy between July and October 2024. The survey explored probiotic prescribing frequency, indications, strains used, dosage, duration, and sources of information. Descriptive statistics and subgroup analyses by years of clinical experience were performed. **Results**: A total of 279 PCPs (response rate: 28%) completed the survey; 66.7% were female, and 77.1% had over 20 years of clinical experience. Probiotics were prescribed primarily to restore microbiota balance (81.1%) and prevent AAD (47.3%). The most common barriers included additional cost (35.1%) and perceived lack of evidence (26.5%). *Lactobacillus rhamnosus* GG (91.8%) and *Saccharomyces boulardii* (41.9%) were the most frequently recommended strains. Daily doses of 5–10 billion CFU were preferred by 44.4% of respondents, with typical durations of 1–2 weeks (40.1%) or one week (31.2%). **Conclusions**: Probiotics are widely used by Italian PCPs during antibiotic therapy, especially for microbiota support and AAD prevention. However, variability in practice underscores the need for clearer, evidence-based guidelines regarding probiotic strain selection, dosing, and treatment duration.

## 1. Introduction

Probiotics are defined as live microorganisms that, when administered in adequate amounts, confer a health benefit on the host. This definition, introduced in 2001 by a joint panel of experts from the Food and Agriculture Organization (FAO) [1] and the World Health Organization (WHO) [2], was reaffirmed in 2014 by the International Scientific Association for Probiotics and Prebiotics (ISAPP) [3]. Probiotics are available in various formulations, including dietary supplements, pharmaceuticals, and food products such as yogurt, juices, and cereals. To be considered a probiotic, the microorganism must be viable at the time of ingestion and administered in sufficient quantities to exert a beneficial effect.

The most commonly used probiotic species belong to the Lactobacillus and Bifidobacterium genera. Frequently studied strains include *Lactobacillus acidophilus*, *L. rhamnosus*, *L. casei*, *L. plantarum*, and *Bifidobacterium longum*, *B. breve*, and *B. bifidum*, among others [4]. In recent years, the European Society for Paediatric Gastroenterology, Hepatology, and Nutrition (ESPGHAN) Working Group on Probiotics and Prebiotics has issued guidelines for the use of probiotics in the prevention and treatment of specific pediatric gastrointestinal disorders. These include acute gastroenteritis, Helicobacter pylori infection, inflammatory bowel disease, functional gastrointestinal disorders, and particularly, antibiotic-associated diarrhea (AAD) [5].

Probiotics are believed to benefit host health by modulating the gut microbiota, influencing intestinal barrier function, and interacting with immune and metabolic pathways [6]. These effects are especially relevant in the context of antibiotic therapy, which is known to disrupt the gut microbial ecosystem. Such disruption can lead to adverse effects, the most common being AAD, a condition defined as diarrhea occurring in association with antibiotic use, in the absence of other identifiable causes [7,8].

In pediatric outpatient settings, antibiotics are among the most frequently prescribed medications in both Italy and other high-income countries [7,8]. However, their use is not always consistent with clinical guidelines, prompting initiatives to promote antibiotic stewardship [9,10]. AAD is a frequent consequence of unnecessary or broad-spectrum antibiotic use, which can promote pathogenic overgrowth (e.g., *Clostridium difficile*) and impair gut microbial diversity [11,12].

Several randomized trials have demonstrated the protective role of probiotics (particularly *Lactobacillus rhamnosus* GG and *Saccharomyces boulardii*) in preventing AAD in children [13,14]. The ESPGHAN Working Group recommends these strains with a strong recommendation and moderate-quality evidence [11]. These probiotics act through various antimicrobial mechanisms, including competitive inhibition, immune modulation, and the production of antimicrobial peptides like bacteriocins and defensins [15]. Though isolated cases of adverse effects, such as systemic infections, have been reported in high-risk patients, probiotics have a strong safety profile for the general population [16,17]. Nevertheless, their clinical efficacy is strain-specific and may vary depending on the formulation, dosage, and duration of treatment [18]. The viability of the preparation (dose), which in turn may depend on the type of matrix, the manufacturing and packaging conditions, the use of encapsulation technology, and the mode of administration—such as environmental interference (microbiome composition or diet)—may determine how effective preparations containing live microorganisms are. Very little systematic data is available for the majority of these issues. Furthermore, the data are scattered since outcomes may vary from application to application and strain to strain. The latter is typically the outcome of Cochrane and systematic reviews. Given this viability concern, standardization, while desirable, is more challenging than for conventional medications. Dietary and regional differences may also be important factors [18].

Despite their widespread use in pediatric practice, especially in conjunction with antibiotics, there are no official national or international recommendations for the routine use of probiotics to prevent AAD. In Italy, all children are assigned a Primary Care Pediatrician (PCP) through the National Healthcare System. PCPs play a central role in prescribing antibiotics and advising on adjunct therapies, including probiotics, thus influencing real-world clinical practice. This study aimed to explore the prescribing behaviors and attitudes of Italian PCPs regarding the use of probiotics during antibiotic therapy. Specifically, the survey investigated indications for use, preferred strains, dosage regimens, and perceived efficacy and safety. The findings are intended to inform future clinical guidelines and support more consistent, evidence-based use of probiotics in pediatric outpatient care.

## 2. Materials and Methods

A cross-sectional survey was conducted using a digital questionnaire consisting of 23 mandatory multiple-choice questions designed to investigate the attitudes and prescribing practices of Italian PCPs regarding the use of probiotics during antibiotic therapy (Appendix A). The questionnaire explored topics such as criteria for antibiotic prescription, frequency and methods of probiotic use, reasons for recommending probiotics during antibiotic treatment, commonly prescribed probiotic strains, perceived efficacy and safety, and sources of information on probiotics. Questions were chosen according to an internal expert panel board (GB, MEC CN, and SE). The study was approved on 12 March 2024 by the Ethics Committee of the University of Parma (UNIPR-2024-PED-2). No personal data were involved (except email addresses), but only an investigation on patient management procedures in the clinical practice.

The survey was administered via Google Forms and distributed to a targeted group of 980 PCPs, randomly selected, representing approximately 14% of the estimated 7000 PCPs practicing in Italy [19]. Invitations to participate were sent through regional professional pediatric societies and direct outreach between 1 July and 30 October 2024. The email invitation included a direct link to the questionnaire. All questions (reported in Appendix A) were required for successful submission, ensuring completeness of the dataset. A follow-up reminder was issued at the end of August 2024. Responses were collected until 30 October 2024. Participation was voluntary and anonymous, and demographic information such as age range and gender was collected. Duplicate entries were prevented by restricting the form to a single submission per respondent. In the survey form, entering an email address was necessary to complete the form. Therefore, no duplicate forms were possible.

In total, 279 PCPs completed the questionnaire, yielding a participation rate of 28%. According to the Italian National Institute of Statistics (ISTAT), there were 6962 registered PCPs in Italy as of 30 May 2024 [19]. Based on this population size, the sample of 279 respondents corresponds to a margin of error of ±2.55% at a 95% confidence interval, providing a statistically representative snapshot of PCPs’ views nationwide. We divided the respondents into two groups according to their years of clinical experience, choosing 20 years as a cut-off. We chose this cutoff given that the majority of the respondents to our survey were older than 55 years old.

Statistical analysis was performed using STATA^®^ software (version 12; StataCorp, College Station, TX, USA). Responses were analyzed descriptively and presented as frequencies and percentages. For multiple-choice questions, percentages were calculated based on the total number of respondents.

## 3. Results

### 3.1. Population Characteristics

Among the 279 PCPs who participated in the survey, the majority were female (186/279, 66.7%), while 93/279 (32.3%) were male. Participants represented a broad age distribution: 10 (3.6%) were aged 25–34 years, 24 (8.6%) aged 35–44, 31 (11.1%) aged 45–54, 112 (40.1%) aged 55–64, and 102 (36.6%) were over 65 years. Responses were received from PCPs across 17 Italian regions, with the highest representation from Puglia (71/279, 25.5%) and Emilia-Romagna (58/279, 20.8%). Most respondents practiced in urban areas (205/279, 73.5%), followed by suburban (53/279, 19.0%) and rural settings (21/279, 7.5%). A substantial portion (215/279, 77.1%) had more than 20 years of clinical experience, suggesting a highly experienced respondent base.

### 3.2. Antibiotic and Probiotic Prescribing Patterns

Regarding the frequency of antibiotic prescription, 55.6% of PCPs reported prescribing antibiotics “occasionally”, 33.7% “frequently”, 9.3% “rarely”, and only 1.4% “very frequently”. Among those prescribing antibiotics occasionally, 60.9% had less than 20 years of experience. A detailed breakdown of antibiotic indications is reported in Table 1.

In terms of probiotic co-prescription, 44.4% of respondents stated they recommended probiotics in more than 75% of antibiotic cases, particularly among those with less than 20 years of experience (51.6%). The remaining pediatricians recommended probiotics in 51–75% of cases (12.5%), 25–50% (14.7%), or less than 25% (19%). Only 9.3% reported never recommending probiotics alongside antibiotics. Factors taken into account by the respondents to our survey when prescribing probiotics during antibiotic therapy are reported in Table 2.

Reasons why the respondents to our survey decided to prescribe probiotics during antibiotic therapy are summarized in Table 3.

### 3.3. Indications and Barriers to Probiotic Use

The most commonly cited reasons for recommending probiotics included restoring intestinal microbiota balance (81.1%) and preventing antibiotic-associated diarrhea (47.3%). These preferences did not differ significantly by years of experience. Less frequently mentioned reasons included general health improvement (20.4%), parental preference (9.0%), and habitual prescribing practices.

Notably, 63.1% of respondents prescribed probiotics whenever antibiotics were indicated, with higher prevalence among those with less than 20 years of practice (73.4% vs. 60.0%, *p* = 0.05). A smaller proportion reserved probiotic use for specific scenarios such as the presence of diarrhea (15.8%), prolonged antibiotic courses (>10 days; 20.1%), or post-treatment only (0.7%). A small percentage (3.9%) never prescribed probiotics.

Barriers to probiotic use included concerns about limited clinical benefit (8.6%), additional cost (35.1%), and lack of supporting scientific evidence (26.5%).

### 3.4. Probiotic Strains, Dosage, and Duration

*Lactobacillus rhamnosus* GG was the most commonly prescribed strain (91.8%), followed by *Saccharomyces boulardii* (41.9%), *Bifidobacterium lactis*, *Lactobacillus acidophilus*, and *Bifidobacterium brevis*, in decreasing order (Figure 1). No significant differences in strain preferences were observed between experience groups.

Most pediatricians prescribed probiotics at daily doses of either 1–5 billion CFU (40.5%) or 5–10 billion CFU (44.4%). Regarding treatment duration after the end of antibiotic therapy, 40.1% recommended a 1–2 week course, while 31.2% advised one week. Longer durations (>2 weeks) were rarely prescribed. Pediatricians with less experience were more likely to prescribe both higher doses and longer durations (Figure 2).

### 3.5. Perceived Efficacy and Safety

Probiotic therapy was considered moderately effective by 42.7% of respondents, with little difference between experience groups. Only 11.1% reported perceiving no efficacy, with more skepticism among those with over 20 years of practice (13.5% vs. 3.1%). The vast majority (91.8%) reported never encountering adverse effects, reinforcing the strong safety profile of probiotics in outpatient care.

### 3.6. Communication with Families and Information Sources

Regarding communication practices, 27.6% of PCPs stated they always discussed the risks and benefits of probiotics with parents, 28.3% often did, and 25.8% sometimes did. A smaller portion rarely (12.2%) or never (6.1%) addressed the topic with families.

The primary sources of information on probiotics were conferences and workshops (68.5%), medical journals (66.3%), and pharmaceutical representatives (55.9%). Fewer participants cited professional associations (10.8%) or colleagues (10%).

### 3.7. Interest in Educational Support and Regional Differences

Interest in further training was high: 73.1% of respondents expressed a desire to attend workshops on probiotic use with antibiotics. A similar proportion (73.1%) supported the development of a national consensus document on this topic, while 21.2% disagreed and 5.7% were undecided.

Geographically, of the 279 participants, 153 were from Northern/Central Italy and 125 from the South/Islands. PCPs in the South more frequently prescribed probiotics in over 75% of antibiotic cases (47.2% vs. 42.5%). Conversely, Northern PCPs were more likely to cite parental preference as a reason for probiotic use (13.1% vs. 4.0%). Full regional comparisons are presented in Table 4.

## 4. Discussion

This national survey provides a comprehensive snapshot of current prescribing practices among Italian PCPs regarding the use of probiotics during antibiotic therapy. In a context where antibiotic resistance and microbiota disruption are major global concerns, these findings contribute valuable real-world data to inform future clinical guidelines and interventions.

Regarding antibiotic prescription, there are both quantitative and qualitative variations in the antibiotic prescriptions given to children across multinational, national, regional, and local levels, as highlighted by Clavenna et al. [20]. The geographical variations in antibiotic usage are influenced by several factors, including healthcare systems that shape drug regulation and the structure of the national pharmaceutical market, physicians’ attitudes (such as uncertainty in diagnosis—particularly in younger patients—or variations in diagnostic labeling, time constraints, or market pressures), as well as sociocultural and economic factors related to patients and their parents (like the patient’s overall health or socioeconomic status) [20].

Our results highlight that probiotics are widely prescribed by Italian PCPs, with 44.4% recommending them in over 75% of antibiotic treatments. The main indications cited were the restoration of intestinal microbiota balance (81.1%) and the prevention of AAD (47.3%), reflecting a general awareness of probiotics’ clinical utility. These findings are consistent with existing evidence supporting the use of specific strains (particularly *Lactobacillus rhamnosus* GG and *Saccharomyces boulardii*) for the prevention of AAD in pediatric patients [11,14,21,22,23,24,25,26].

However, the study also reveals substantial variability in prescribing patterns, particularly regarding treatment duration, dosage, and selection criteria. While most PCPs recommended probiotics for 7 to 14 days after the end of antibiotic treatment, some suggested shorter or longer durations, reflecting the absence of standardized national guidelines. Dosages ranged mostly between 1 and 10 billion CFU/day, again without a clear consensus. These inconsistencies emphasize the need for more precise, evidence-based protocols to guide probiotic use in routine pediatric care.

A particularly interesting finding is the generational divide in prescribing behavior. PCPs with less than 20 years of experience were significantly more likely to prescribe probiotics systematically during antibiotic therapy (73.4% versus 60%), suggesting greater familiarity with recent microbiome research. In contrast, 10.7% of senior PCPs with over 20 years of experience reported that they never recommend probiotics. To the best of our knowledge, this is the first survey to provide this kind of result. This discrepancy highlights the importance of continuous medical education and the integration of updated clinical evidence into practice, regardless of years in service. Interestingly, no statistically significant regional difference was reported. Current evidence does not support the indiscriminate use of probiotics in all cases of antibiotic administration. Rather, as outlined in the recent ESPGHAN Position Paper on probiotic use [5], their implementation should be selectively considered based on clinical judgment. This decision-making process should incorporate variables such as the specific antimicrobial agent employed, the duration of therapy, and the patient’s baseline health status. However, a recent Cochrane review found no significant adverse events associated with probiotic administration in hospitalized and outpatient pediatric populations. Observational data from intensive care and neonatal settings have reported serious adverse outcomes in critically ill or immunocompromised children. These events were predominantly observed in patients with predisposing factors, including the presence of central venous catheters and clinical conditions conducive to bacterial or fungal translocation [14].

Economic barriers were also evident. Over one-third of participants cited the additional cost of probiotics as a reason against their routine use. Given the lack of reimbursement in many healthcare settings, cost remains a significant limitation to widespread adoption, particularly in lower-income or underserved populations. Policymakers should consider strategies to ensure broader access to evidence-based probiotic therapies.

Although the study provides timely and nationally representative data, several limitations must be acknowledged. The response rate, while in line with similar voluntary surveys, was 28%, raising the possibility of selection bias, as those more engaged with the topic may have been more likely to participate. Furthermore, the data are based on self-reported practices, which may be subject to recall bias or social desirability bias, potentially overestimating probiotic usage rates. The cross-sectional design does not allow for an assessment of actual patient outcomes, such as the incidence or reduction of AAD following probiotic use. Additionally, although regional differences were captured, the study did not explore in detail the cultural, organizational, or socioeconomic factors that may influence prescribing behavior. Another important limitation is the absence of specific data on vulnerable populations, such as immunocompromised children, where probiotic safety and efficacy remain less well defined. Furthermore, an additional limitation is the absence of data on the specific types of antibiotics co-prescribed when probiotics were recommended. This information was not included in the original survey and thus precludes analysis of whether probiotic use varied according to the antibiotic’s spectrum, duration, or appropriateness of prescription. As AAD risk is influenced by the antibiotic class and its impact on gut microbiota, understanding the interplay between specific antibiotics and probiotic prescribing practices would have offered more nuanced insights. For instance, it remains unclear whether probiotics were more commonly recommended when broad-spectrum or potentially inappropriate antibiotics were used. Future research should integrate data on antibiotic types and regimens to better elucidate the rationale behind probiotic use and to support targeted, evidence-based recommendations. Despite these limitations, this study has several important strengths. It offers one of the few large-scale evaluations of probiotic use in pediatric primary care in Europe and includes a diverse sample across age groups, clinical experience levels, and geographic regions. The structured design of the questionnaire and its mandatory-response format ensured data completeness and internal consistency. Moreover, by stratifying responses according to years of practice, the survey identifies meaningful generational trends that could inform targeted educational strategies.

## 5. Conclusions

Probiotics are widely adopted in pediatric outpatient care, primarily to support gut microbiota balance and prevent AAD, particularly in the context of prolonged or high-impact antibiotic therapy. However, significant variability in clinical practice underscores the need for clearer, evidence-based guidance regarding their indications, optimal strains, dosing regimens, and duration of use. Future research should prioritize identifying the most effective probiotic formulations for AAD prevention, evaluating their cost-effectiveness, and assessing the long-term impact on the pediatric microbiome. Additionally, the development of standardized national and international guidelines is essential to harmonize prescribing practices and ensure consistent, high-quality care.

Efforts to improve pediatricians’ knowledge of probiotic use (including indications, safety profiles, and strain-specific efficacy) should be integrated into continuing medical education. Furthermore, incorporating patient and parental perspectives into future studies is crucial, as understanding their attitudes and expectations may influence adherence and the perceived benefit of probiotic interventions. Addressing these research and practice gaps will support a more rational and effective use of probiotics in pediatric care, ultimately enhancing patient outcomes and reinforcing antibiotic stewardship.

## Figures and Tables

**Figure 1 antibiotics-14-00577-f001:**
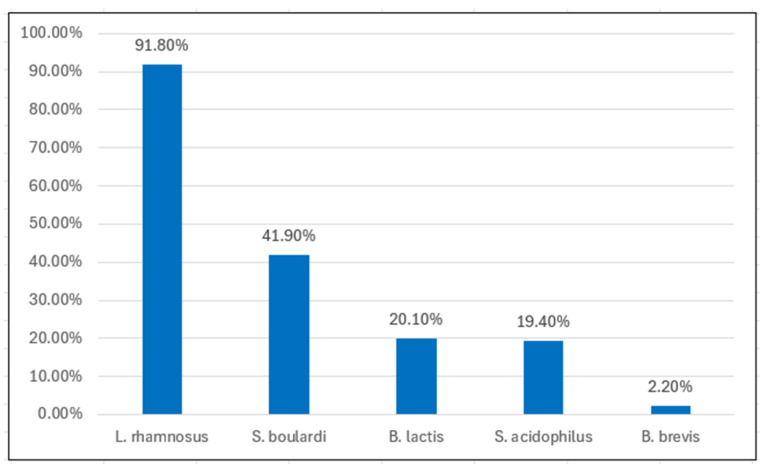
Most frequently prescribed probiotic strains among Primary Care Pediatricians (PCPs) during antibiotic therapy.

**Figure 2 antibiotics-14-00577-f002:**
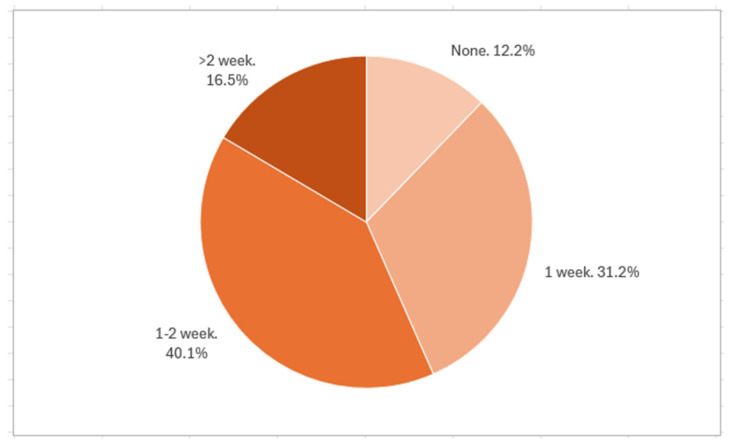
Duration of probiotic therapy after the end of antibiotic therapy prescribed by Primary Care Pediatricians (PCPs).

**Table 1 antibiotics-14-00577-t001:** Common conditions treated with antibiotics by Primary Care Pediatricians (PCPs) participating in the survey, stratified by years of clinical practice (<20 years vs. ≥20 years).

Common Conditions for Which You Prescribe Antibiotics (More than One Answer Is Possible)			
Clinical condition	PCPs with <20 years of practice	PCPs with ≥20 years of practice	Total PCPs
Respiratory tract infections	57/64 (89.1%)	172/215 (80%)	229/279 (82.1%)
Ear infections	55/64 (85.9%)	172/215 (80%)	227/279 (81.4%)
Skin infections	26/64 (40.6%)	110/215 (51.2%)	136/279 (48.8%)
Urinary tract infections	46/64 (71.9%)	159/215 (74%)	205/279 (73.5%)
Other medical conditions	64/64 (100%)	215/215 (100%)	279/279 (100%)

**Table 2 antibiotics-14-00577-t002:** Conditions related to probiotics use during antibiotic therapy by Primary Care Pediatricians (PCPs) participating in the survey.

Conditions Related to Probiotics Use During Antibiotic Therapy			
Clinical condition	PCPs with < 20 years of practice	PCPs with ≥20 years of practice	Total PCPs
If the child has diarrhea at the time of the antibiotic prescription	12/64 (18.8%)	32/215 (14.9%)	44/279 (15.8%)
When I prescribe a therapy lasting more than 10 days	10/64 (15.6%)	46/215 (21.4%)	56/279 (20.1%)
Every time I prescribe antibiotic therapy	47/64 (73.4%)	129/215 (60%)	176/279 (63.1%)
After antibiotic therapy in the presence of diarrhea	0/64	1/215 (0.5%)	1/279 (0.4%)
Depending on the type of antibiotic	1/64 (1.6)	0	1/279 (0.4%)
Always at the end of antibiotic therapy	0/64	2/215 (0.9%)	2/279 (0.7%)
Never	1/64 (1.6)	10/215 (4.7%)	11/279 (3.9%)
Always	0/64	3/215 (1.4%)	3/279 (1.1%)

**Table 3 antibiotics-14-00577-t003:** Reasons for recommending probiotics among Primary Pediatricians (PCPs) participating in the survey, stratified by years of clinical practice (<20 years vs. over 20 years).

Main Reasons to Prescribe Probiotics During Antibiotic Therapy			
Reason	PCPs with < 20 years of practice	PCPs with ≥20 years of practice	Total PCPs
To restore gut microbiota balance	58/64 (82.8%)	173/215 (80.5%)	226/279 (81%)
To improve the patient’s general health	15/64 (23.4%)	39/215 (18.1%)	54/279 (19.4%)
To prevent antibiotic-associated diarrhea	35/64 (54.7%)	97/215 (45.1%)	132/279 (47.3%)
Repeated or prolonged antibiotic therapies	0 (0%)	1/215 (0.5%)	1/279 (0.4%)
I do not prescribe probiotics	1/64 (1.6%)	11/215 (5.1%)	12/279 (4.3%)
Parental preference	6/64 (9.4%)	19/215 (8.8%)	25/279 (9%)

**Table 4 antibiotics-14-00577-t004:** North/Central vs. South/Islands differences.

Do You Recommend Probiotics When Prescribing Antibiotics?	North	South	Total
**Never**	18 (11.8%)	7 (5.6%)	25 (9%)
In less than 25% of cases	31 (20.3%)	22 (17.6%)	53 (19%)
In 25–50% of cases	20 (13.1%)	21 (16.8%)	41 (14.7%)
In 51–75% of cases	19 (12.4%)	16 (12.8%)	35 (12.5%)
In more than 75% of cases	65 (42.5%)	59 (47.2%)	124(44,4%)
**If yes, what are the main reasons for recommending probiotics? (select all that apply)**			
Restore the balance of the intestinal microbiota	121 (79.1%)	105 (84%)	226 (81%)
Improve the general health of the patient	28 (18.3%)	26 (20.8%)	54 (19.4%)
**Prevent** antibiotic-associated diarrhea	71 (46.4%)	61 (48.8%)	132(47.3%)
Repeated or prolonged antibiotic therapy	1 (0.7%)	(0)	1 (0.4%)
Parental preference	20 (13.1%)	5 (4%)	25 (9%)
I do not prescribe probiotics	7 (4.6%)	4 (3.2%)	11 (3.9%)
**Where do you mainly get information about probiotics? (select all that apply)**			
Medical journals	104 (68%)	80 (64%)	184(65.9%)
Pharmaceutical representatives	88 (57.5%)	68 (54.4%)	156(55.9%)
Professional associations	15 (9.8%)	14 (11.2%)	29 (10.4%)
Conferences and workshops	96 (62.8%)	94 (75.2%)	190(68.1%)
Colleagues	15 (9.8%)	13 (10.4%)	28 (10%)
**Would you be interested in attending a workshop on using probiotics with antibiotics?**			
Maybe	32 (20.9%)	21 (16.8%)	53 (19%)
No	35 (22.9%)	15 (12%)	50 (17.9%)
Yes	86 (56.2%)	89 (71.2%)	175(62.7%)

## Data Availability

All the data are included in the manuscript.

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
