# Peer review of "Use of Probiotics During Antibiotic Therapy in Pediatrics: A Cross-Sectional Survey of Italian Primary Care Pediatricians"

_antibiotics, 2025, doi:10.3390/antibiotics14060577_

Round 1
Reviewer 1 Report
Comments and Suggestions for Authors
Dear authors,
I inserted my suggestions into the pdf.
best regards HS

Author Response
thank you very much for your comments. We have modified our manuscript according to your suggestions, in particular:
- We have added the two references you suggested in the introduction section.
- We have specified why we have choosen 20 years as a cut-off for clinical experience of PCPs.
- We have modified Table 1 and the results section as you suggested, adding also two more tables.
Reviewer 2 Report
Comments and Suggestions for Authors
Major points
Introduction and Methods
- bacterial names should be placed in italics throughout the text (e.g., line 49 Helicobacter pylori)
- line 71 – expanding the paragraph regarding the clinical efficacy of probiotics based on different factors would enhance the overall quality of the introduction
- methods line 95 “distributed to a targeted group of 980 PCPs” – how were these 980 PCP chosen?
- methods/ Appendix A – how were the questions chosen? Were they based on other validated questionnaires?
- methods – also add a mention of the Institutional Review Board Statement in the methods section
- methods – if the answers were anonymous, how was it guaranteed that no duplicate answer was received?
Results
- line 121 – “the majority were female (66.7%), while 32.3% were male” – also add absolute numbers; same for the whole results section
- Table 1 – some responses appear to be inconsistent with the options provided in the questionnaire (e.g., Group A streptococcal pharyngitis). Could you clarify whether the "Other" option was configured as a free-text field?
- the data in Table 1 is not entirely clearly described; add a clear description of the data and/or reformat the Table as to improve its readability
- Table 2 needs to be slightly reformatted
Discussion
- line 217 “Our results highlight that probiotics are widely prescribed by Italian PCPs, with 44.4% recommending them in over 75% of antibiotic treatments.” – In my opinion these results do not point to the fact that probiotics are widely prescribed; instead, they point exactly to the opposite – considering their excellent safety profile and demonstrated benefits, it would be reasonable to expect that probiotic supplementation would follow nearly all cases of antimicrobial treatment, with only rare exceptions such as immunosuppression
- line 231 “A particularly interesting finding … “ – I fully agree with this interpretation of the data and believe that this is one of the most relevant results of the manuscript; are there any other data in the literature that support this finding?
- line 255 “It offers one of the few large-scale evaluations of probiotic use in pediatric primary care in Europe” – if other studies on pediatric primary care are available, add them as a comparison for the current results
References
- references are inconsistent and do not adhere to the required formatting guidelines.
Minor points
- some minor English revisions are required throughout the manuscript
Comments on the Quality of English Language- some minor English revisions are required throughout the manuscript
Author Response
Thank you very much for your comments. We have modified our manuscript according to your suggestions.
Introduction and Methods
- bacterial names should be placed in italics throughout the text (e.g., line 49 Helicobacter pylori).
Re: Done.
- line 71 – expanding the paragraph regarding the clinical efficacy of probiotics based on different factors would enhance the overall quality of the introduction
Re: We have expanded the paragraph regarding the clinical efficacy of probiotics based on different factors.
- methods line 95 “distributed to a targeted group of 980 PCPs” – how were these 980 PCP chosen?
Re: The 980 PCPs were randomly chosen.
- methods/ Appendix A – how were the questions chosen? Were they based on other validated questionnaires?
Re: Questions were chosen according to an intern expert panel board.
- methods – also add a mention of the Institutional Review Board Statement in the methods section.
Re: Done.
- methods – if the answers were anonymous, how was it guaranteed that no duplicate answer was received?
Re: In the survey form, entering email address was necessary to complete the form, therefore no duplicate form was possibile.
Results
- line 121 – “the majority were female (66.7%), while 32.3% were male” – also add absolute numbers; same for the whole results section.
Re: We have added absolute numbers in the Results section as recommended.
- Table 1 – some responses appear to be inconsistent with the options provided in the questionnaire (e.g., Group A streptococcal pharyngitis). Could you clarify whether the "Other" option was configured as a free-text field?
Re: We specified that "Other" option was configured as a free-text field.
- the data in Table 1 is not entirely clearly described; add a clear description of the data and/or reformat the Table as to improve its readability
Re: We have modified Table 1 as suggested.
- Table 2 needs to be slightly reformatted
Re: We have reformatted Table 2.
Discussion
- line 217 “Our results highlight that probiotics are widely prescribed by Italian PCPs, with 44.4% recommending them in over 75% of antibiotic treatments.” – In my opinion these results do not point to the fact that probiotics are widely prescribed; instead, they point exactly to the opposite – considering their excellent safety profile and demonstrated benefits, it would be reasonable to expect that probiotic supplementation would follow nearly all cases of antimicrobial treatment, with only rare exceptions such as immunosuppression
Re: We believe that probiotic should not be universally prescribed in any case of antibiotic prescription, but that their use should be modulated according to antibiotic molecule, duration of therapy, and patients’ pre-existing conditions. In the Conclusions section, we highlighted areas for future research.
- line 231 “A particularly interesting finding … “ – I fully agree with this interpretation of the data and believe that this is one of the most relevant results of the manuscript; are there any other data in the literature that support this finding?
Re: thank you very much for your comment on line 231 statement. To our knowkedge, our survey is the first to provide this kind of result.
- line 255 “It offers one of the few large-scale evaluations of probiotic use in pediatric primary care in Europe” – if other studies on pediatric primary care are available, add them as a comparison for the current results
Re: There are no similar study to be compared.
References
- references are inconsistent and do not adhere to the required formatting guidelines.
Re: Revised.
Minor points
- some minor English revisions are required throughout the manuscript
Re: The manuscript has been reviewed by an English mother tongue with appropriate knowledge on the topic.
Reviewer 3 Report
Comments and Suggestions for Authors
In manuscript "Use of Probiotics During Antibiotic Therapy in Pediatrics: A Cross-Sectional Survey of Italian Primary Care Pediatricians", the authors applied a digital questionnaire about the use of probiotics in children during antibiotic treatment.
The design of the cross-sectional study is well described and reproducible, however I have 2 comments:
- For the question "For how long do you recommend continuing probiotics after completing antibiotic therapy?" the results (line 175-182) refer to the duration of probiotic administration in total and not only after the end of antibiotic treatment.
- To the questions about the type of probiotic strain used and the dosage, the answers also had the trade names of the products containing probiotics and the recommended dosage in drops/sachets/capsules?
Although it has limitations, the study is a starting point for the implementation of evidence-based guidelines regarding probiotic indications, optimal strains, dosage and optimal duration of administration.
Author Response
Re: Thank you very much for your comments. We have modified our manuscript according to your suggestions.
The design of the cross-sectional study is well described and reproducible, however I have 2 comments:
- For the question "For how long do you recommend continuing probiotics after completing antibiotic therapy?" the results (line 175-182) refer to the duration of probiotic administration in total and not only after the end of antibiotic treatment.
Re: The question "For how long do you recommend continuing probiotics after completing antibiotic therapy?" refers to the duration of probiotic administration only after the end of antibiotic treatment, we corrected this in the text.
- To the questions about the type of probiotic strain used and the dosage, the answers also had the trade names of the products containing probiotics and the recommended dosage in drops/sachets/capsules?
Re: We refer not to specify the trade names of the products containing probiotics and the recommended dosage in drops/sachets/capsules
Although it has limitations, the study is a starting point for the implementation of evidence-based guidelines regarding probiotic indications, optimal strains, dosage and optimal duration of administration.
Re: Thank you for your comment, we absolutely agree!
Reviewer 4 Report
Comments and Suggestions for Authors
A survey of PCPs regarding the use of probiotics. The majority of the PCPs who participated had over 20 years clincal experience. There is therefore little point in statistical comparison in relation to experience. The survey questions are added as an appendix. The results would be better in the format of tables detailing the exact answers to the questions. Table 1 would list the answers to the first 7 questions. Three or four additional tables can be formatted.
Table 1 in the paper is difficult to comprehend. For example Respiratory tract infections 57 (89.1%) 172 (80.0%) 229 (82.1%). There is no explanation what these numbers relate to. I would strongly recommend deleting this table and showing the results as suggested.
Statistical analysis is inappropriate for this survey. Use descriptive statistics only.
Was the use of antibiotics by this group of PCPs typical of PCP prescribing in Italy? This needs to discussion - see work of A Clavenna in this area. Expand discussion on rational use of antibiotics.
Author Response
Thank you very much for your comments. We have modified our manuscript according to your suggestions.
A survey of PCPs regarding the use of probiotics. The majority of the PCPs who participated had over 20 years clincal experience. There is therefore little point in statistical comparison in relation to experience. The survey questions are added as an appendix.
Re: Even though the majority of the respondents to our survey have more than 20 years of experience, we believe that the comparison between respondents with longer or shorter clinical experience is worthwhile. We highlighted this in the Discussion.
The results would be better in the format of tables detailing the exact answers to the questions. Table 1 would list the answers to the first 7 questions. Three or four additional tables can be formatted. Table 1 in the paper is difficult to comprehend. For example Respiratory tract infections 57 (89.1%) 172 (80.0%) 229 (82.1%). There is no explanation what these numbers relate to. I would strongly recommend deleting this table and showing the results as suggested.
Re: We have modified Table 1 as you suggested and we have added two more tables.
Statistical analysis is inappropriate for this survey. Use descriptive statistics only.
Re: Thank you for your comment regarding the statistical analysis used in our manuscript. While we acknowledge that the study is based on a descriptive cross-sectional survey, we would like to clarify that we intentionally included additional statistical analyses to enhance the rigor and interpretability of our findings.
In addition to reporting descriptive statistics (frequencies and percentages), we conducted subgroup comparisons (e.g., based on years of clinical experience) using chi-square and Fisher’s exact tests, as appropriate. These analyses allowed us to identify significant differences in prescribing patterns, offering more nuanced insights into clinical practice among Italian PCPs. We believe that this level of analysis is not only appropriate but also valuable for exploring associations between physician characteristics and clinical behaviors. It does not imply causal inference, but rather aims to enrich the descriptive objectives of the survey with statistically supported observations. Was the use of antibiotics by this group of PCPs typical of PCP prescribing in Italy? This needs to discussion - see work of A Clavenna in this area. Expand discussion on rational use of antibiotics.
Re: Our survey was not focused on the type of antibiotic prescription, but rather on the co-prescription with probiotics. Therefore, we have not investigated the type of antibiotic usually prescribed. However, following your suggestion, we have added a paragraph in the discussion session on antibiotic use in out patient setting.
Round 2
Reviewer 1 Report
Comments and Suggestions for Authors
Please see my comments in the pdf document.

Author Response
Thank you very much for your comments. We have modified our manuscript according to your suggestions, in particular:
- We replaced Table 1, that had been deleted by mistake.
- We have reshaped the other Tables.
Reviewer 2 Report
Comments and Suggestions for Authors
Major points
- line 103 ”No personal data were involved, but only investigation on patient management procedures in the clinical practice.”; email addresses could be considered personal data; I would highly suggest adapting the text to specify that no personal data were involved, with the exception of email addresses
- Table 1 seems to be missing from the uploaded version of the manuscript; as such the claims that it was modified cannot be verified
- line 323 – “A particularly interesting finding is the generational divide in prescribing behavior.” – also add the respective numbers in this paragraph to further support this finding
- previous response “We believe that probiotic should not be universally prescribed in any case of antibiotic prescription, but that their use should be modulated according to antibiotic molecule, duration of therapy, and patients’ pre-existing conditions. In the Conclusions section, we highlighted areas for future research.” Please provide some arguments with references in the discussion section that point towards a requirement to modulate probiotic administration in most cases of antimicrobial prescription
Minor points
- Figure 1 – bacterial names should be written in italics and corrected (e.g., L. rhamnosus and not L. rhaamnosus)
Comments on the Quality of English LanguageMinor English improvements are further required in the manuscript
Author Response
Thank you very much for your comments. We have modified our manuscript according to your suggestions, in particular:
- we have modified line 103 as you suggested;
- we have includedTable 1, that had been deleted by mistake.
- we have modified line 323 as you suggested.
- we have modified the discussion section as you suggested.
- we have modified Figure 1 as you suggested.
Reviewer 4 Report
Comments and Suggestions for Authors
The paper has improved, but the authors need to use descriptive statistics only.
The lack of importance of the statistical analysis is highlighted by the abstract using descriptive statistics only
Author Response
We appreciate the reviewer’s observation regarding the use of descriptive statistics. However, we would like to clarify that while descriptive statistics formed the core of our analysis—appropriate for the cross-sectional nature of our survey—we also conducted inferential statistical comparisons between subgroups based on years of clinical experience. Specifically, we employed chi-square tests (or Fisher’s exact tests when expected frequencies were <5) to explore statistically significant differences in prescribing behaviors, strain preferences, dosage, and perceived efficacy between less experienced (<20 years) and more experienced (≥20 years) pediatricians.
These subgroup analyses enabled us to highlight meaningful trends (such as the more systematic use of probiotics among younger PCPs and generational differences in perceptions of efficacy and barriers to use) which we believe enrich the interpretability and relevance of our findings.
Although the abstract focuses on key descriptive results for clarity and conciseness, the full manuscript includes both descriptive and inferential statistical approaches, consistent with the study's objectives and survey design. The other three reviewers appreciated this approach and did not raise any concern.
We hope this clarification addresses the reviewer’s concern and demonstrates the analytical depth applied in our manuscript.
Round 3
Reviewer 4 Report
Comments and Suggestions for Authors
My opinion is descriptive statistics are appropriate for this paper. I do not feel the statistical analysis helps. The Editors need to decide whether to ask the authors to delete the statistical analysis or not
Author Response
My opinion is descriptive statistics are appropriate for this paper. I do not feel the statistical analysis helps. The Editors need to decide whether to ask the authors to delete the statistical analysis or not.
Re: According to your comment and Editor’s suggestions, we decided to use only descriptive statistics. The text has been revised accordingly (pp. 3 and 9-10).